# Transient Unfolding and Long-Range Interactions in Viral BCL2 M11 Enable Binding to the BECN1 BH3 Domain

**DOI:** 10.3390/biom10091308

**Published:** 2020-09-11

**Authors:** Arvind Ramanathan, Akash Parvatikar, Srinivas C. Chennubhotla, Yang Mei, Sangita C. Sinha

**Affiliations:** 1Data Science and Learning Division, Argonne National Laboratory, Lemont, IL 60439, USA; 2Consortium for Advanced Science and Engineering, University of Chicago, Chicago, IL 60637, USA; 3Department of Computational and Systems Biology, University of Pittsburgh, PA 15260, USA; akash.parvatikar@gmail.com; 4Department of Chemistry and Biochemistry, North Dakota State University, Fargo, ND 58108, USA; ymei@Wistar.org

**Keywords:** BCL2, BECN1, intrinsic disorder, transient folding, machine learning

## Abstract

Viral BCL2 proteins (vBCL2s) help to sustain chronic infection of host proteins to inhibit apoptosis and autophagy. However, details of conformational changes in vBCL2s that enable binding to BH3Ds remain unknown. Using all-atom, multiple microsecond-long molecular dynamic simulations (totaling 17 μs) of the murine γ-herpesvirus 68 vBCL2 (M11), and statistical inference techniques, we show that regions of M11 transiently unfold and refold upon binding of the BH3D. Further, we show that this partial unfolding/refolding within M11 is mediated by a network of hydrophobic interactions, which includes residues that are 10 Å away from the BH3D binding cleft. We experimentally validate the role of these hydrophobic interactions by quantifying the impact of mutating these residues on binding to the Beclin1/BECN1 BH3D, demonstrating that these mutations adversely affect both protein stability and binding. To our knowledge, this is the first study detailing the binding-associated conformational changes and presence of long-range interactions within vBCL2s.

## 1. Introduction

Human and other vertebrate cells encode B-cell lymphoma 2 proteins (cBCL2s), which are important regulators of homeostatic pathways, such as apoptosis and autophagy [1,2,3,4]. Oncogenic viruses, such as Epstein–Barr virus (EBV), Kaposi’s sarcoma-associated virus (KSHV), and the murine-γ-herpesvirus 68 (γ-HV68), express homologs of cBCL2s [5]. These viral BCL2s (vBCL2s) bind to cellular pro-apoptotic or pro-autophagic proteins to mediate the inhibition of apoptosis and autophagy, thereby supporting the survival of infected host cells [6,7,8,9] and preventing viruses from being destroyed [10]. VBCL2s are important for reactivation from latency and oncogenic transformation [11]. Currently, there are no drugs to treat γ-herpesvirus infections; however, vBCL2s are attractive targets, as blocking the function of these proteins can prevent virus survival and replication altogether [12]. For example, recent studies have shown that biologicals that selectively bind to vBCL2s can prevent these proteins from inhibiting autophagy [13,14,15]. However, designing vBCL2 inhibitors is challenging, because of the high degree of sequence and structural variation amongst vBCL2s and a lack of understanding of the binding-associated conformational changes in vBCL2s.

A key challenge in elucidating the atomic details of how vBCL2s bind to their target proteins stems from the conformational flexibility of the binding partners [16]. In particular, vBCL2s bind to the BCL2 homology 3 domains (BH3Ds) of pro-apoptotic proteins [17,18], as well as to a BH3D within the essential autophagy effector, BECN1 (BCL2-interacting coiled-coiled protein) [19,20], leading to the inhibition of apoptosis and autophagy, respectively [17,18,19,20,21,22]. These BH3Ds are intrinsically disordered domains when not bound to a partner [16], but bind as helices to a conserved surface cleft between $\alpha_{2}$ and $\alpha_{3}$ of vBCL2s (Figure 1) and cBCL2s. Each cBCL2 binds with high affinity to subsets of BH3Ds, but not to others [23]. In contrast, vBCL2s appear to be more promiscuous, binding a wider range of BH3Ds, but to none with high affinity [19]. Therefore, in spite of the structural similarity between cBCL2s and vBCL2s, there are clear differences in the BH3D binding mechanisms to these homologs. In particular, within cBCL2s, while it has been proposed that allosteric interactions, i.e., interactions between residues distant from the binding cleft can affect BH3D binding [24,25,26,27,28], the mechanism by which this allosteric modulation of function is achieved remains unknown. Further, it is also unknown whether vBCL2s exhibit similar allosteric interactions [29].

In previous publications, we have described the direct interactions between residues lining the binding cleft of of a typical vBCL2, M11, and the BECN1 BH3D in substantial detail, including quantifying the impact of mutating these interface residues on binding of the two protein [13,19]. In this study, we investigated binding-associated conformational transitions of a typical vBCL2, M11, to quantitatively elucidate how vBCL2s bind to BH3Ds. M11 is perhaps the best characterized vBCL2, as structures of both, the unliganded (apo-M11; Figure 1A) and the BECN1 BH3D-bound state (holo-M11; Figure 1B) are available. Interestingly, despite the binding of a large, helical BECN1 BH3D, the overall root-mean squared deviation (RMSD) between the apo-M11 and holo-M11 is only about 1.8 Å. The conformational changes are chiefly localized to the BH3D binding groove and involve the displacement of tightly packed hydrophobic residues (Tyr52-Tyr56-Tyr60) (Figure 1C,D).

Given the localized structural change observed in the apo- and holo-M11, we asked whether the apo-M11 can transiently access conformations that resemble holo-M11. Using microsecond time-scale molecular dynamics (MD) simulations of apo- and holo-M11, and our previously developed trajectory analysis tools that seek higher-order statistical signatures in time-dependent structural changes, we show that, despite the relatively low RMSD between the apo- and holo-M11 [30,31,32,33], the dominant modes of motions of the apo-M11 simulations do not sample conformations that are similar to the holo-M11. Further, we show that the opening and closing of the BH3D binding site is intrinsically coupled to the partial unfolding of helices α1, α2 and α5. We find that the tight hydrophobic interaction between Tyr52-Tyr56-Tyr60 from α2 or α3 helices acts as a conformational gate, preventing the unfolding of α2/α3 and opening of the binding site. In addition to these residues, our computational analyses of M11 also reveals a more extensive network of hydrophobic interactions (some of which are more than 10 Å from the BH3D binding groove) that play a central role in the opening/closing of the binding groove. We show that perturbing this network of interactions reduces M11’s binding affinity to BECN1 BH3D by more than a 100-fold. Together, our studies, for the first time, suggest that, like the cBCL2s, allosteric interactions in vBCL2s, like M11, can modulate the binding of BH3Ds. A better understanding of these long-range interactions opens up opportunities to design novel allosteric regulators of vBCL2 function.

## 2. Methods

### 2.1. System Preparation and Conformational Sampling

Molecular dynamics (MD) simulations were carried out for both the *apo-* and *holo-*forms of Bcl-2. We used three structures: (1) apo form of Bcl-2 from the NMR ensemble (PDB: 2ABO), referred to as 2ABO-apo; (2) holo form of Bcl-2 in complex with BECN1 (PDB: 3DVU), referred to as 3DVU-holo, and (3) apo-M11 initialized without the BECN1 bound to the binding site, referred to as 3DVU-apo. Although the two structures are identical in terms of their primary sequence, we noted the presence of two additional C-terminal residues, namely Glu135-Asp136 in the 2ABO-apo structure as well as a N-terminal residue Lys5 in the 3DVU-apo/holo structures. For structure preparation, these additional residues were modeled into the respective simulations. However, these residues were not considered for further analyses, since we were primarily interested in understanding the core structural components that interact with BECN1 BH3D, including $\alpha_{2-6}$ and L1–L4. Each of the systems was processed using the AMBER 14 suite of molecular modeling software, with the FF14SB [34] force-field. Standard amino-acid residues were used to build the protein structures and the hydrogen atoms were added according to the protonation state for each amino acid determined at pH 7.0. For each of the systems, neutralizing charges were also added in order to enable a stable system for simulation. Point mutations were generated using AMBER 14 suite of tools based on specific amino acid substitutions. The proteins were placed in a rectangular box of TIP3P waters, such that the distance between the boundary of the box and protein was at least 12 Å. Each of the systems was equilibrated using a procedure outlined in previous work [35]. Briefly, the water molecules were first minimized while using steepest descent (500 steps), followed by a conjugate gradient minimization until the root-mean squared (RMS) error was less than 0.25 kcal/mol·Å. Following this, protein atoms were minimized in order to release any bad contacts in the crystal structures. A small MD simulation (25 ps) with a gradual increase in temperature to 300 K was carried out, followed by a constant pressure simulation (25 ps), so that the unrestrained water molecules could occupy vacuous regions within the protein. Five additional runs of equilibration were performed using constant volume with each step consisting of an energy minimization (threshold of 0.0001 kcal/mol·Å) and a 5 ps MD run. The solute (protein and counter-ions) were restrained in these runs. A harmonic restraint of 100 kcal/mol·Å${}^{2}$ was applied in the first equilibration run; this harmonic restraint was reduced by half in successive steps, and no restraints were applied in the final equilibration run. After using a temperature ramp to gradually heat the system to a temperature of 300 K, a constant pressure simulation of 25 ps was carried out in order to fill in any remaining voids in the protein with the solvent.

The production runs were performed using AMBER version 14.0, using the constant number of particles, constant volume, and energy (NVE) ensemble, with particle mesh Ewald (PME) technique for electrostatics and a 10 Å cut-off for Lennard–Jones interactions. The SHAKE algorithm [36] was used to restrict motions of all covalently bound hydrogen atoms. Simulations were performed at 1 atm pressure and at 300 K temperature. The data from the simulations were stored every 0.05 ns, resulting in a total of 20,000 snapshots for 1 μs sampling for each of the three systems. Production runs were carried out in five replicates for each of the three systems (as well as the mutant systems), resulting in a total of 17 μs of aggregate sampling.

### 2.2. Anharmonic Conformational Analysis (ANCA)

The partial unfolding of α2 in Bcl2 simulations suggests that Bcl2 undergoes significant anharmonic structural changes in the course of the simulation. Based on the observation that the thermal motions of atoms appear to produce positional fluctuations that have significant higher-order moments [32,37,38], we hypothesize that anharmonic motions may relate to protein function. To quantify the anharmonic time-dependent conformational changes, we have previously introduced an approach, called *anharmonic conformational analysis* (ANCA) [30,39]. ANCA uses fourth-order statistics to describe the atomic fluctuations and summarizes the internal motions using a small number of dominant anharmonic modes. We have successfully demonstrated this approach in the context of both molecular recognition (ubiquitin and lysozyme) and enzyme catalysis (human cyclophilin A) [30,32,33]. An emergent property of ANCA is that anharmonicity enables the discovery of energetically homogeneous conformational substates.

### 2.3. Temporal Decorrelation of Fourth-Order on Time-Delayed Cumulant Matrices

In the interest of resolving spatial and temporal anharmonic dependencies in the molecular simulation trajectories, we have designed the TD4 module that performs joint diagonalization of time-delayed cumulant matrices (a tensor of fourth-order time-delayed statistics signifying kurtosis). TD4 is the counterpart of SD4, where fourth-order spatial correlations are minimized, implying zero time lag.

Conceptually, the assumption we make is that a molecular simulation trajectory is a linear combination of independent, anharmonically fluctuating protein motions. To discover these anharmonic motions, we borrow a technique from signal processing literature, called Blind Source Separation (BSS) [40], which attempts to extract or unmix independent non-Gaussian sources from signal mixtures with Gaussian noise. In order to facilitate the extraction of anharmonic modes of motion of the fourth-order, the trajectory data $X_{orig}$∈$R^{3N\times t}$, where 3*N* represents *(x,y,z)* coordinates from individual atom selections and *t* represents conformations is decorrelated for second-order dependencies both spatially and temporally by transforming it through the modules of SD2 and TD2. SD2 module removes dominant second order spatial correlations by computing a spatial covariance matrix and performing principal component analysis (PCA). The function diagonalizes covariance matrix to obtain the projection matrix $Y=B^{T}X_{orig}$(m×t), where *m* is subspace dimensionality and *B*(3N×m) are the dominant eigenvectors. Consequently, the TD2 module removes dominant second order temporal correlations by computing a time-delayed (specified by a lag time τ) covariance matrix and performing PCA. A matrix *Z* is obtained by projecting the spatially resolved data matrix *Y* onto the dominant eigenvectors $B_{TD2}$. The matrix *Z* then undergoes transformations to retrieve mutually independent signals by obtaining a separating matrix *W*∈$R^{m\times 3N}$.

Algorithmically, the method of unmixing temporally correlated signals of fourth-order can be viewed as a symmetric eigenvalue problem of a generalized cumulant matrix $Q_{i\;j}$. As a measure of statistical independence, we will consider the ’diagonality’ of a set of cumulant matrices. The cumulant matrices are generated in a low-dimensional subspace denoted by *m*, which is the best guess for the most compact summary of the fourth-order statistics. The subspace dimensionality can be adjusted by examining the inflection points in the cumulative variance plots generated from SD2 module.

In order to generate the cumulant matrices, a time-lagged covariance matrix is defined by:(1)$$R_{z}\left(\tau \right)=E\left\{ZZ_{\tau }^{T}\right\},$$
where *Z* ∈$R^{m\times t}$ is second-order spatially and temporally resolved molecular simulation data, τ is time delay, and $Z_{\tau }=Z\left(t-\tau \right)$ is the time-lagged version of *Z*. A fourth-order cumulant matrix $Q_{i\;j}$ of this data matrix *Z* is defined by:(2)$$Q_{i\;j}=E\left\{ZZ^{T}Z_{\tau }^{T}Z_{\tau }\right\}-E\left\{ZZ^{T}\right\}\mathrm{tr}\;E\left\{Z_{\tau }Z_{\tau }^{T}\right\}-2E\left\{ZZ_{\tau }^{T}\right\}E\left\{Z_{\tau }Z^{T}\right\},$$
where $Q_{i\;j}$∈$R^{m\times m}$ computes a time-lagged cumulant matrix. The possibility of computational errors, such as round-off errors, can destroy the symmetricity of the cumulant matrix which is restored by performing:(3)$$Q_{i\;j}=\frac{1}{2}\left[Q_{i\;j}+Q_{i\;j}^{T}\right].$$

A time-lagged cumulant tensor Q∈$R^{m\times (m\times k)}$, where *k* = [m×(m+1)]/2 is defined for the storage of cumulant matrices computed by the symmetric $Q_{i\;j}$ matrix. Joint diagonalization of these time-lagged cumulant matrices reduces fourth-order temporal dependencies, leading to anharmonic modes of motion of the trajectory data. This is done through Jacobi’s iterative method of finding solution to a system of linear equations. In particular, the method uses successive transformations in order to calculate diagonal elements of the cumulant tensor by decimating off-diagonal elements with each iteration. The spatio-temporally decorrelated matrix of fourth-order is computed by obtaining:(4)$$Z_{TD4}=WX_{orig},$$
where *W* attempts to separate sources from signal mixture $X_{orig}$ by finding directions, such that projections onto these directions have maximum statistical independence. The computed parameter $Z_{TD4}$ is a fourth-order spatially and temporally resolved matrix.

### 2.4. Identifying Conformational Sub-States

Observe that neighboring conformers in QAA space have biophysically relevant coordinates (see Section 3.2), and this coherence is an emergent property of the QAA representation. Based on this observation, we hypothesize that the nearest neighbors in the QAA space can be hierarchically clustered to form dynamically and kinetically related meta-stable states.

To this end, we consider each frame in the trajectory as a node in an undirected graph that is constructed in the three-dimensional QAA space. We connect each node to a small number of its nearest Euclidean neighbors. The edges weights are binary (either 0 or 1) denoting either the presence or absence of an edge. We then cluster this network using a hierarchical Markov diffusion framework that we have proposed previously [31,39,41]. First, we initiate a Markov chain to propagate on the network and identify a set of putative cluster centers. The total number of clusters is automatically determined by the algorithm with the rule that every node in the network has some Markov probability of transitioning into at least one of the putative cluster. Second, a Markov transition matrix is built while using this reduced representation based on the principle that Markov chains initiated on both the fine scale and coarse scale representations of the network should reach stationary distributions simultaneously. This principle, in turn, helps to build a hierarchal representation of the network and promotes the formation of meta-stable clusters in the data.

We expect that fine-grained hierarchy levels will produce many small clusters containing close neighbors in the QAA space; that is, within each such cluster most members will be drawn from the same, narrow time-window. As Markov diffusion progresses (fine-grained to coarse-grained), conformers that are more distant neighbors will be connected by edges in the diffused network and will, therefore, be assigned to the same cluster. Thus, the hierarchical clustering can highlight dynamical connections between conformers at different timescales.

### 2.5. Experimental Methods

***Peptide synthesis, protein expression and purification.*** The Beclin 1 BH3D peptide was chemically synthesized as described previously [13]. WT M11 was expressed while using the pET21(d+)-γHV68 M11 plasmid used in our previous work (1). Site-directed mutagenesis (Agilent), using appropriate primers, was used to generate pET21(d+)-γHV68 M11 mutants of F17A, F48A, Y56A, A63K, F79A, Y97A, L103A, M119A, and L128A with the primers, respectively. WT and mutated M11 plasmids were expressed and purified, as described previously [19]. Purified WT and mutant M11 protein samples were concentrated to 0.1 mM, except for the M11 F17A mutant, which did not yield sufficient protein.

***Isothermal Titration Calorimetry (ITC).*** ITC was performed using a Low Volume Nano ITC (TA instrument). The ITC buffer for all experiments was 50 mM HEPES pH 7.5, 150 mM NaCl and 2 mM βME. For all ITC experiments, the Beclin 1 BH3D peptide and various purified M11 constructs were loaded into separate dialysis cassettes and co-dialyzed against 2 L ITC buffer overnight. The ITC experiments were performed at 14 ${}^{\circ }$C with 25 injections of 2 μL each BH3D titrating into M11 protein. The resulting data were plotted and analyzed by NanoAnalyze Software (TA instrument) with an independent model.

## 3. Results

We carried out MD simulations using three systems in order to gain insights into BH3D-binding associated conformational changes in M11: (1) 2ABO-apo: which refers to the simulation of apo-M11 initiated from the unliganded M11 NMR ensemble (PDB ID 2ABO [42]); (2) 3DVU-apo: which refers to the simulation started from the holo-M11 X-ray crystal structure (PDB ID 3DVU [19] chain A), but with the BECN1 BH3D removed from the binding cleft; and, (3) 3DVU-holo: which refers to the simulation of holo M11, with the BECN1 BH3D bound to the M11 binding cleft (PDB ID 3DVU chains A and B). In this paper, apo-M11 simulations collectively refer to both 2ABO-apo and 3DVU-apo simulations; when individual differences are observed and summarized, we explicitly refer to the individual simulations. On the other hand, the holo-M11 only refers to the 3DVU-holo simulation. Each of these systems was equilibrated and production runs were carried out for a total of 1 μs per system using AMBER 14 (see Methods section and Appendix A) [43]. A comparison of the chemical shifts from the simulations and the experiments yielded strong agreement (average ρ = 0.9; Appendix A). For the holo-M11 simulations, we compared the overall root-mean squared fluctuations (RMSFs) against the experimental B-factors (Appendix A) and observed strong correlations, which indicated that both of these simulations were stable and reasonably capture experimental observables in terms of overall atomic fluctuations (at the time-scales simulated).

### 3.1. Partial Unfolding of M11-α2 Facilitates Opening/Closing of BH3D Binding Site

***RMSF in M11 simulations indicate regions far away from the binding cleft are impacted by BECN1 BH3D binding.*** To understand how the M11 is affected by BH3D binding, we examined the conformational fluctuations quantified by the RMSF per-residue for the different M11 simulations (Figure 2A,B). The RMSF plots show BECN1 BH3D binding not only affects residues localized to the binding cleft of vBCL2, but also residues farther from the binding site. Regions that show large fluctuations, labeled according to their location in vBCL2, are shown in Figure 2A. For clarity, these regions are also mapped on the M11 structure in Figure 2B. Regions with the largest changes in RMSF, including α2, L2, α3 (highlighted in red) and loop L3 (highlighted in green) in Figure 2A, are proximal to the binding cleft (Figure 2B). The helix α3 that unfolds and gets displaced (Figure 2B) provides the necessary space to accommodate the BECN1 BH3D. In comparison, the flexible loops L1 (highlighted in blue) and L4 (highlighted in orange) in Figure 2B, are both distal to the binding site (average distance from the binding cleft center of about 16 Å) display smaller, yet significant, fluctuations. For the 3DVU-apo simulation, the RMSF profiles indicate larger fluctuations around the binding site as a consequence of removing the BH3D (Figure 2A).

These observations led us to hypothesize that α2 partially unfolds when the BECN1 BH3D is not bound to the binding cleft. Therefore, we examined the time-evolution of the secondary structure content while using the DSSP program [44,45]. As illustrated in Appendix A, as compared to the 3DVU-holo simulations, the 2ABO-apo simulations exhibit significant secondary structure transitions within α2. In particular, the span of α2 comprising of residues 33–37 undergo transient helix-to-coil transitions in the 2ABO-apo simulations. This can be mainly attributed to the nature of fluctuations within the hydrophobic core within the protein consisting of residues 88–91 ($\alpha_{5}$) that interact with residues 41–48 in α2 (see Figure 2C). We observe this partial unfolding across our *apo-*state simulations. The possible mechanism of this transition is further explored in the next subsection, where we posit that these partial unfolding and refolding motions enable the protein’s binding site to adapt to its substrates, including BECN1 BH3 domain. Further, while the span of residues 58–63 at the C-terminal end of α2 remains stable in a helical conformation in the 2ABO-apo simulation, the same region undergoes partial unfolding in the 3DVU-apo state simulations. These observations indicate that intrinsic partial unfolding and refolding of α2 may play a role in enabling BH3D binding.

***Partial unfolding in M11 α2 reveals re-arrangements in the hydrophobic contacts within α1 and α5.*** Given the inherent conformational changes in α2, we examined the contacts between residues comprising this helix and other helices of M11. We define a contact to exist if the C${}^{\alpha }$ or C${}^{\beta }$ atom’s distance between residue *i* (from α2) and residue *j* (from other helices in M11) is less than or equal to 6 Å. This definition has been used in other studies, including the construction of elastic network models [46,47] and it has been used to quantify folding/unfolding processes within proteins [48,49]. We then defined the lifetime of a contact as the fraction of the simulation time for which the distance between the residues was less than or equal to 6 Å. Based on these definitions, we found that residues of α2 directly contact residues in both α1 and α5, but not in either α4 or α6. An average contact map (Figure 2C,D) displays the contacts between residues of α2 and residues of α1 and α5.

As illustrated in Figure 2C–F, there is a clear distinction in the lifetimes of contacts observed between the apo- and holo-M11 simulations for the interactions between α1–α2 (Figure 2C) versus α2 and α5 (Figure 2D). Notably, the number of transient interactions (contact lifetimes of <0.3) in the apo-M11 (2ABO-apo, 3DVU-apo) are higher than the holo-M11 (3DVU-holo), which clearly shows the presence of more stable interactions (contact lifetimes of >0.5). Consequently, the contacts in the holo form are more stable than the apo state of vBCL2. This is also evident from the histograms of contact lifetimes that are shown in Figure 2E,F, where the holo-M11 simulation (red solid line) shows a significant increase in stable interactions as compared to the apo-M11 simulations. Although there is only a marginal increase in the contact lifetimes between α1–α2, these changes nevertheless further indicate that α2 is more stable in their holo-M11 than in the apo-M11 simulations. Within the holo-M11 simulations, α2 remains stable (at μs time-scales), and displays a stronger hydrophobic packing pattern, indicating that the BH3D binding induces significant changes to the conformational fluctuations within α2. All of these hydrophobic interactions are at least 10 Å from the BH3 domain binding site, indicating that these unfolding/refolding observed in α2 is likely to play a role in BH3D binding.

The partial unfolding observed in α2 and its fluctuations in the contact lifetimes relative to α5 led us to further hypothesize that this unfolding may play a role in the opening/closing of the BH3D binding site. To examine this hypothesis further, we extracted all of the conformations, where α2 retains a partially unfolded structure (the final 0.4 μs of the 2ABO-apo simulation; Appendix A) and computed the volume of the binding cleft in M11 for the apo- simulations. As shown in Appendix A, the volume of the binding cleft is significantly larger upon partial unfolding of α2 in the 2ABO-apo simulations, indicating that the unfolding observed in α2 is coupled with the motions in the binding cleft to accommodate a large ligand, such as BECN1 (or other BH3Ds).

### 3.2. Structural Intermediates of the Apo- to Holo-M11 Show Transient Unfolding of α2

***Quasi anharmonic analysis (QAA) reveals how conformational changes in the M11-BH3D binding site are coupled to motions across the entire protein.*** A consequence of unfolding observed in the apo-state simulations is that it may facilitate the opening of the binding groove within M11 and, therefore, allow for the placement or anchoring of BECN1 BH3D. To examine this hypothesis, we used QAA (described in Methods section) to partition the microsecond long apo-M11 simulations into conformational substates. QAA characterizes higher-order statistical signatures in atomic fluctuations to organize the conformational landscape of a protein into a small number of conformational substates. Partitioning the apo-M11 into substates allows the description of the apo-simulations in terms of a small number of reaction coordinates that can then be used to understand how the transitions in the landscape are driven, i.e., understand the intrinsic and potentially complex coupling that may exist between conformational changes in α2 and the binding groove in M11. We applied QAA on the C${}^{\alpha }$ atoms; the top 50 QAA basis vectors describes nearly 90% of the overall variance observed in the simulations.

We project the trajectory from the apo-M11 simulations onto the top three QAA modes ($\gamma_{1-3}$) as shown in Figure 3A. To show that the QAA representation is biophysically meaningful, we paint each conformer in this space by its RMSD value with respect to the starting structure in the simulation. The conformers in this reduced dimensional space appear to cluster into putative substates sharing structural similarities. We also project the holo-M11 simulations onto the top three QAA modes (Figure 3B) and observe that the holo-M11 conformers (painted as orange dots) span putative substates II, III, and IV. The conformational transitions that enable apo-M11 to access putative substates II, III and IV, can be visualized as movies, as shown in Figure 3C–E, respectively. As the movies progress from the apo-M11 state (marked I in Figure 3A) to the holo-M11 putative substates (II, III, IV), in each mode, one can observe fluctuations in the loop regions, namely L1, L2, L3 and L4, and partial unfolding of α2 and α3 region, expanding the BH3D binding site. These coupled motions observed from QAA depict how partial unfolding of α2 facilitates the opening/ closing of the BH3D binding site. Further, the motions that are encoded by the top three QAA modes show that conformational fluctuations much farther away from the BH3D binding site may play a role in opening/closing the site.

***Intermediate substates of apo- to holo-M11 transition reveal concerted motions between M11 binding groove and α1 and α5.*** The three QAA modes ($\gamma_{1-3}$) from the apo-M11 simulations depict coupled motions in M11 that result in opening or closing of the binding pocket (see Appendix A), which, in turn, enable the binding of the BECN1 BH3D. To better understand the intrinsic coupling between α2, L2, α3, L3, and α4 of M11 (Figure 2A,B), we apply a graph-theoretic clustering technique to seek out intermediates that may mediate the structural transition from apo- to holo-M11. We construct a nearest neighborhood graph, where each conformer (regardless of its apo- or holo-origin) is connected to a small number of its neighbors based on their similarity in the QAA space spanned by the QAA modes ($\gamma_{1-3}$). A Markov chain is initiated on this graph to generate a clustering hierarchy. Starting from the lowest level of the hierarchy, the clustering procedure successively groups together conformers into larger substates based on their structural and dynamical similarities in the QAA space, until only the two apparent states, the apo- and holo-M11, remain. As one progresses through the different levels of the hierarchy, this procedure allows for one to successively examine local to global conformational changes that result in the apo- to holo- state transition in M11. Here, we provide evidence for one possible hypothesis, where the unfolding of α2 and subsequent transitions enable M11 to bind to BH3D.

We arrange the 47 conformational substates found at a coarse level of the hierarchy to progress from apo-enriched states to holo-enriched states (Figure 4A). The matrix entries denote the similarity, or *affinity*, between the substates, in both their structural and dynamical features, as captured by QAA. The affinity matrix is automatically computed at each level of the hierarchy (see Methods for more details). In particular, we observe that there are 14 substates that are exclusively populated by apo- state conformers, 12 substates that are populated by holo-state conformers, and 21 substates that show the presence of both apo- and holo- state conformers. Substates S${}_{1}$ and S${}_{47}$ exclusively represent the apo- and holo- states, respectively (Figure 4A). Five substates, namely S${}_{4}$, S${}_{9}$, S${}_{12}$, S${}_{16}$, and S${}_{20}$ mediate the transitions between the apo- and holo- states based on a simple measure of the proportion of apo- and holo- state conformers in these substates.

The five intermediate substates can be visualized as a part of a network connecting the apo-M11 to the holo-M11 substates (Figure 4B). Each of the intermediate states connect with both apo- and holo- state simulations. Within the intermediate states (S${}_{4}$, S${}_{9}$, S${}_{16}$ and S${}_{20}$), a series of local unfolding motions within α2 (red helix) occur along the N-terminal end propagating towards the C-terminal end. The largest motions observed across residues 33–50 (Figure 4C). Further, the cartoon representations of substates (indicated by the red arrows) also depict how the unfolding motions within α2 propagate. Initially, the motions are local starting at the C-terminal end of α2, closer to the binding cleft and gradually propagate towards the N-terminal end of α2. However, the interconnections between the substates imply multiple unfolding motions can take place (perhaps simultaneously), indicating that the protein can use multiple pathways to sample the holo-state conformations. These unfolding transitions are coupled to other changes that are highlighted in different colors in Figure 4C in the context of opening up the binding cleft, by displacing α3 and α4–α5 loop regions.

These conformational changes subsequently perturb the hydrophobic packing between α2 and α5 (green and cyan regions in Figure 4C) and between α2 and α1 (red region in Figure 4C), leading to a collective displacement of α5 and α4 within the binding cleft. The substates also observed exhibit complementary motions within L3 (yellow color in Figure 4C) and L4 regions (indicated by blue in Figure 4C). Whether these complementary fluctuations observed are specific to a particular BH3D is not examined within this paper; however, we observe several sites within BECN1 BH3D that undergo complementary motions resulting in opening/closing of the binding site. Thus, the network representation allows for us to identify dominant pathways that eventually lead the apo-state simulation to access holo-state conformational substates in the context of opening of the BECN1 BH3D binding groove.

### 3.3. Perturbing Hydrophobic Interactions Farther from M11 Binding Cleft Impact BECN1 BH3D Binding

Based on the analysis of the intermediate states in Figure 4, we identified thirteen residues, Phe17, Ile41, Phe48, Tyr52, Tyr56, Tyr60, Ala63, Leu74, Phe79, Phe92, Leu103, Met119, and Leu128 that facilitate the apo- to holo-M11 transitions in our MD simulations. These residues are not localized to one region, but dispersed throughout the protein. We can also observe that these residues form an inter-connected network of hydrophobic interactions (depicted in Figure 5 center panel), connecting the BH3D binding groove with other regions in M11.

***Mutation of Phe48 drastically reduces binding BECN1 BH3D.*** Phe48 is about 9 Å from the center of the BECN1 BH3D binding site and packs against the hydrophobic core of M11. It interacts with Tyr52 (Figure 5A) in both the apo- and holo-M11 simulations. The 2ABO-apo and 3DVU-holo simulations show remarkably stable interactions between Tyr52 and Phe48, as shown in Figure 5A. However, in the 3DVU-apo simulations (green lines in Figure 5A), this interaction is dynamic and can be attributed to the destabilization of the local hydrophobic interactions between residues Tyr52, Ile103, and Ile41 (within the binding site) once the BECN1 BH3D domain is removed. Notably, the 3DVU-apo simulations do sample distances between the two residues similar to the 2ABO-apo simulations, as shown in Figure 5A; however, the hydrophobic interactions are more dynamic than in the 3DVU-holo state indicating that locally destabilizing these interactions can potentially alter BECN1 BH3D binding. Based on the tight interaction between Phe48 and Tyr52 in the apo- and holo-M11 simulations (Figure 5A), and the fact that the C-terminal end of α2 (including Phe48) undergoes partial unfolding in substates S${}_{12}$ and S${}_{16}$ (Figure 4C), we hypothesized that mutating this residue can have a significant impact on the binding of BECN1 BH3D. Consistent with these predictions, quantification of binding affinity by ITC (Table 1) demonstrates that mutating Phe48 to Ala essentially abrogates binding of the BECN1 BH3D, displaying about 100-fold weaker affinity.

These interactions indicate that the α3 region starts to unfold as a consequence of removing BECN1 BH3D from the binding site.

**Mutation of Tyr56 renders α2 in apo- state more flexible but does not affect the BECN1 BH3D bound state.** A second residue that is in the direct vicinity of the binding groove is the residue Tyr56, which is located in between Tyr52 and Tyr60. As observed from Figure 5B, the distance variations of Tyr52–Tyr56 (top) and Tyr56–Tyr60 (bottom) in the 2ABO-apo (black) and 3DVU-holo (green) simulations indicate relatively stable distances. In the 3DVU-apo simulations, however, the Tyr52–Tyr56 interaction is more dynamic and undergoes conformational rearrangements pulling Tyr52 closer towards Tyr56. Thus, we posited that Tyr56 may act as a hydrophobic gate, stabilizing the apo-state of the protein.

In order to better understand the role of Tyr56, and to test whether it acts as a hydrophobic gate by stabilizing the α3 region and preventing it from unfolding, we mutated Tyr56 to an glycine in silico both in the apo- and holo-M11 and observed how this mutation affected the overall conformational flexibility of the protein. We note that this is a more drastic modification than in the experiments; however, this was carried out to elucidate the role Tyr56 plays within the binding site. As shown in Appendix A, the mutation at residue position Tyr56Gly entails a significant change in the atomic RMSF, making α2 more flexible, as compared to the wild type apo-simulations. Given that residues comprising of α2 are quite far away from this mutation site (nearly 12 Å away), the effect of this mutation involves a significant portion of the protein, indicating that the mutation can destabilize the apo-M11. Further, slower motions along the simulation show that this disruption of the packing ends up significantly displacing α2 closer to the α4 region, thus significantly narrowing the binding pocket. Surprisingly, in the holo-state simulations, the mutation does not alter the bound BECN1 BH3D, as can be observed from Appendix A, since the RMSF profiles before and after the mutation are almost identical. Therefore, the Tyr56 mutation does not disrupt any of the key interactions between BECN1 and M11, and the peptide remains bound in the binding site (at the time-scale of these simulations). These in silico analyses are supported by our experimental data (Table 1), which show that mutating Tyr56 only minimally reduces the binding of BECN1 BH3D.

***Mutation of Ala63 to Lys decreases BH3D binding.*** Ala63 forms stable hydrophobic interactions with Tyr60 and Leu74. In the apo-M11 simulations, Ala63 interacts with Leu74 (Figure 5C top panel). In the holo-M11 simulations (Figure 5C bottom panel), however, Ala63 is buried within the binding site, and interacts with both the bound BECN1 BH3D (side chain of Arg115 and the buried Leu116). As a consequence, it interacts with the hydrophobic residues in BECN1 BH3D as well as Tyr60. Given the significant positional relocation it undergoes between the apo- and holo-M11, and the fact that the intermediate states indicate a partial unfolding of the helical turn formed by Tyr60, we hypothesized that the hydrophobic interaction may be important for stabilizing the bound BECN1 BH3D within the binding cleft of the M11 protein. Therefore, in our experiments, we mutated Ala63 to the positively-charged residue Lys, which would disrupt the hydrophobic interactions between α3 and BECN1 BH3D. This mutation does indeed diminish binding of the BH3D, although the effect is not as drastic as for the Phe48 mutations (Table 1).

***Mutation of Phe17 to Ala destabilizes M11.*** Phe17 is a residue that is almost 20 Å away from the closest part of the binding groove of vBCL2. It packs tightly against residues Ile41 (from α2) and Phe92 (from $\alpha_{5}$). A cursory observation of the distances between the C${}^{\gamma }$ atoms of the individual residues does not show significant differences between the 2ABO-apo, 3DVU-apo and 3DVU-holo simulations. However, when we examined the $\chi_{2}$ angles for Phe17 over the course of the simulations (Figure 5D), we observed that the 3DVU-holo and 3DVU-apo simulations sample distinct conformations (Figure 5D) of the $\chi_{2}$ not observed in the 2ABO-apo simulations. These additional conformations sampled in the 3DVU-apo simulations show that the rotation of the side-chain impacts the movement of the entire hydrophobic core of the protein allowing for the binding groove to expand and accommodate a large substrate, such as BECN1 BH3D. Further, in the apo-M11 (both 2ABO-apo and 3DVU-apo) simulations, we observe that the side-chains of Phe17, Ile41, and Phe92 are correlated in their motions, keeping the hydrophobic interactions intact within M11. Mutating Phe17 to Ala (as shown in Table 1) resulted in very poor expression of the mutant M11, suggesting that the mutant was unstable and/or misfolded.

We also assessed the impact of mutating additional residues: Tyr79, Tyr97, Leu103, Met119, and Leu128, and each of these mutations resulted in lower binding affinities to BECN1 BH3D, indicating that perturbing the interactions can impact M11’s overall binding mechanism (Appendix A).

## 4. Discussion

Targeting the apoptotic and autophagic machinery within the cell is an important research area for the development of therapeutics for human health issues, ranging from cancer, cardiovascular, neurological diseases resulting from aberrant protein aggregation, and infectious diseases [50,51]. Within oncogenic viruses, targeting anti-apoptotic/anti-autophagic vBCL2s offers strategic benefits, since drugs can specifically target a critical part of the viral life-cycle, namely viral reactivation from latency and proliferation [29,52]. However, elucidating the mechanistic details of how vBCL2s interact with a variety of BH3Ds of pro-apoptotic and autophagy effectors has remained a long-standing challenge. Specifically, the details of the conformational changes in vBCL2s that accompany binding to large BH3Ds have never been elucidated. Such studies are critical not only to understand the mechanism of binding, but also for the identification of potential allosteric sites, which can be targeted by the development of selective therapeutics that target vBCL2s, but not their human host cell homologs [29].

In this paper, we report how the viral BCL2 M11 binds to the BH3D of an essential autophagy effector BECN1 using both all-atom MD simulations and mutational studies. The simulations and subsequent analyses of the data provide initial hypotheses for which residues in M11 play an important role in binding, and the experiments subsequently validate the hypotheses from our simulations. In particular, we show: (1) how the apo-M11 partially unfolds to adapt its binding cleft to accommodate a large helical domain such as the BH3D and (2) how this partial unfolding/refolding of M11 is enabled by a network of hydrophobic interactions that are farther away from the binding cleft and yet significantly impact BH3D binding.

The substate hierarchy reveals a series of conformational transitions between apo- and holo-M11 protein (Figure 4). By utilizing the anharmonic modes of motion that encode the opening and closing of the BH3D binding site, we discovered that only a small number of substates facilitate transitions between the apo- and holo- states. These coupled motions from the top three anharmonic modes disrupt the hydrophobic interactions in the α2 region, and lead to the partial unfolding required for BH3D binding.

A closer examination of these motions further reveals that the perturbations in α2, α3 and α4 helices of M11 are mediated by a small number of hydrophobic residues. This network consists of Leu44, Phe48, and Tyr52 from α2, Tyr56 and Tyr60 from α3, Trp67, Leu74, and Phe75 from α4, and Gly86, Arg87, and Tyr97 from α5. We have already shown how mutations to Tyr60, Leu74, Gly86, and Arg87 can have a significant impact on BECN1 binding [19]. In this study, we show that Tyr60 and Leu74 are part of a larger network of hydrophobic interactions that undergo concerted conformational changes to enable binding of BH3Ds. Further, this network includes residues farther away from the binding cleft, such as Phe17, Phe48, and Ala63. We also validate the role of these residues in this network by demonstrating that mutating these residues negatively impacts BH3D binding (Table 1), and in some cases M11 stability.

Based on our observations, we propose that the partial unfolding within M11 is functionally relevant for modulating the extent to which the BH3D binding site can expand/contract. In particular, the extent to which the α2 region unfolds either in the N-terminal or its C-terminal region (Figure 4) modulates the extent to which α3 and α4 helices are displaced within the BH3D binding site. Thus, given that vBCL2s, like M11, can bind to a variety of BH3Ds (albeit with lower affinities), the partial unfolding of α2 region enables sufficient structural plasticity to accommodate different BH3Ds. Significantly, this partial unfolding mechanism observed in M11 for binding BH3Ds is similar to other BCL2 members, including the cellular Bcl-xL, which undergoes partial unfolding in a structurally equivalent α-helix to bind PUMA BH3D [28]. Similarly, a survey of the variation in available Mcl-1 and Bcl-xL crystal structures also implicates moderate flexibility as being critical for facilitating interactions between BH3Ds and BCL2 proteins [53].

To our knowledge, this is the first study on a prototypical vBCL2 that shows that allosteric interactions can mediate partial unfolding within the binding cleft to accommodate the BECN1 BH3D. This proposed mechanism is complementary to the network proposed by Ionescu and colleagues [27], where a network of interactions primarily consisting of charged residues was proposed to be important for cellular BCL2s to bind and interact with BH3Ds. Although we did not implicate charge transfer as a pre-requisite for vBCL2 binding, our analysis of the allosteric network suggests that flanking the allosteric interactions are several charged residues whose interactions may also play a role in modulating the binding mechanism of this protein. Further studies will be required to understand the role of these allosteric interactions, but, from cellular BCL2s, such as Bcl-xL, it is clear that PUMA BH3D interactions can disrupt p53 binding [28]. These allosteric sites can potentially offer insights into alternate binding sites, where small molecules can be targeted [29].

We note that the set of simulations carried out in this study included 2ABO-apo, 3DVU-holo, and 3DVU-apo states. Potentially, one could also have included the BECN1-BH3D into the 2ABO-apo state within the same simulation box in order to monitor the conformational states as well as mechanisms that ar involved in the binding process. Although such studies for elucidating the binding mechanisms of other protein–protein interactions are common [54,55,56,57], we note that the size of the simulation box and solvation conditions could have a pronounced effect in terms of drawing scientific conclusions. Further, a recent study by Gapsys and de Groot [58] also highlighted the importance of considering box width and volume for accurate sampling and estimating kinetics and other relevant parameters. We plan to extend our simulations with some of our adaptive sampling techniques as part of a future study given that the conformational changes for the BH3D in the presence of the viral BCL2 takes place in millisecond timescales [59].

Many important questions still remain that we propose to address through future studies. We wouod like to determine if allosteric interactions are found only in M11 or whether they are common to other vBCL2 homologs. Strikingly, although vBCL2s are poorly conserved (Appendix A), the network of hydrophobic interactions appears to be conserved across other vBCL2 structures. This is in spite of the fact that the individual residues in specific positions can vary. This provides preliminary evidence suggesting that similar transitions that are mediated by analogous hydrophobic networks are involved in facilitating binding of BH3Ds to other vBCL2s. Future detailed simulations and experimental studies across other vBCL2 systems will provide quantitative evidence delineating the mechanims of these interactions. We also plan to study whether this allosteric network contributes to the specificity of binding to different BH3Ds and whether BH3D binding be modulated by phosphorylation at distant vBCL2 residues. BECN1 is a conformationally-flexible, multi-domain protein [60] and therefore it becomes necessary to understand how other BECN1 domains impact binding to M11 and whether the proposed network of hydrophobic interactions in M11 play a role in interacting with additional BECN1 domains. Further, it is also important to understand if this hydrophobic network differentially modulates binding of diverse BH3D-containing proteins. The combination of simulations and experiments used here provide a novel and powerful framework to address these questions and also identify allosteric sites within vBCL2s that can be targeted by new therapeutics.

## Figures and Tables

**Figure 1 biomolecules-10-01308-f001:**
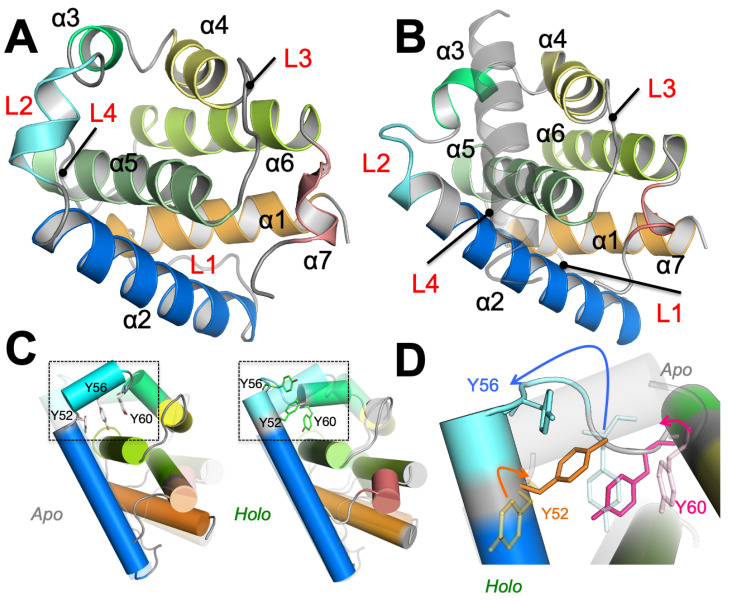
**Conformational changes in apo- and holo-M11**. M11 helices are delineated by different colors. (**A**) Apo-M11 (PDB: 2ABO) and (**B**) Holo-M11 (PDB: 3DVU). The BECN1 BH3D is displayed as a gray transparent helix. The flexible loops between the different α helices: α1–α2 (L1), α2–α3 (L2), α4–α5 (L3) and α5–α6 (L4) are labelled in red. (**C**) The apo- and holo-M11 when superimposed have an root mean squared deviation of 1.8 Å. We separately depict the apo- and holo-M11 structures. Notably, structural overlap depicts differences in loop areas farther away from the binding groove (highlighted by a dotted rectangle. (**D**) A close-up of the dotted rectangle highlighted in (**C**) shows the differences between the apo- and holo-M11 structures in the vicinity of the BH3D binding site. Notably, Tyr52, Tyr56, and Tyr60 undergo significant conformational changes that are highlighted by the colored arrows.

**Figure 2 biomolecules-10-01308-f002:**
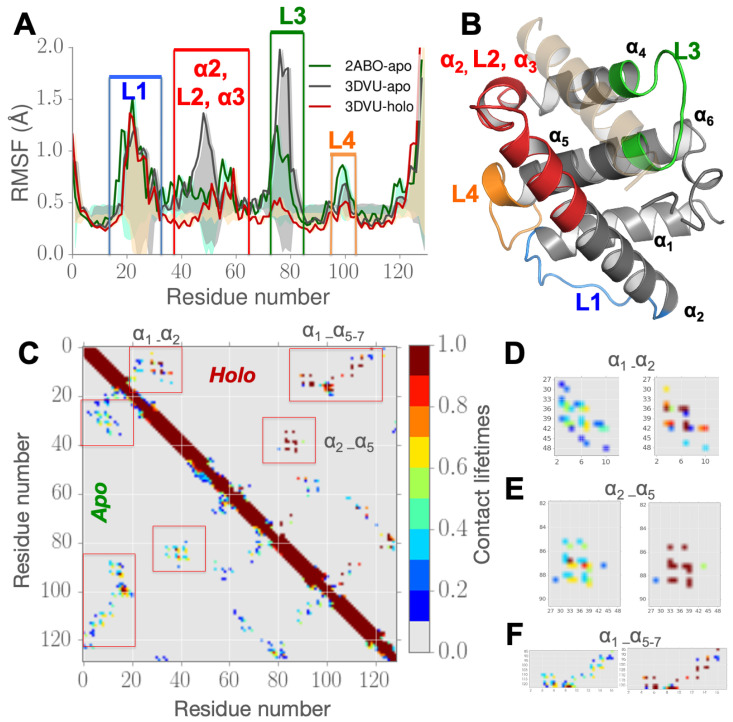
**RMSFs and contact re-arrangements from microsecond long MD simulations in vBCL2 within apo- and holo-M11 simulations show partial unfolding and refolding of α2**. (**A**) RMSF plots for the apo-M11 simulations (2ABO-apo shown in green solid line and 3DVU-apo shown in gray solid line) and holo-M11 simulations (initated from 3DVU structure; red solid line) showing RMSF values plotted per residue (at the corresponding C${}^{\alpha }$ atoms). (**B**) Four regions that show significant differences in RMSF upon binding in (**A**) are highlighted on the 3DVU-holo structure. The bound BH3 domain (BECN1 from 3DVU structure) is highlighted in orange transparent cartoon. (**C**) Lifetime of contacts between the apo (lower triangle) and holo (upper triangle) to demonstrate the contrast in fluctuations between the two states, and to understand the conformational changes after BH3D binding. A higher lifetime indicates higher stability and lower lifetime indicates no contact or less stable interactions. (**D**) Distinct areas of changes between apo- and holo- states calculated from the simulations can be observed across highlighted rectangles, which are zoomed in an inset for residues lining (**D**) α1 (6–18) and α2 (28–48); (**E**) α2 and α5 (88–98); and, (**F**) $\alpha_{1}$, $\alpha_{5-7}$ (90–110). Our simulations posit a significant increase in the stability of long-lived contacts after BECN1 BH3D binding, especially across regions farther away from the binding site.

**Figure 3 biomolecules-10-01308-f003:**
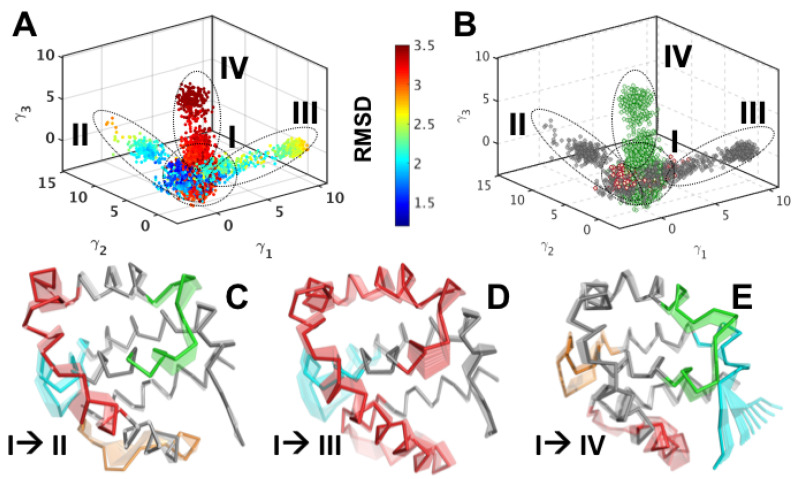
**Quasi-anharmonic analysis of apo-M11 simulations indicates how coupled motions in α2 control the opening and closing of the BH3 binding site.** (**A**) Projection of the 1 μs apo-M11 simulations along the top three anharmonic modes ($\gamma_{1-3}$) shows the presence of at least 4 putative substates in the landscape that have distinct structural (and energetic) features. Each conformation in the projection plot is colored accorting to the RMSD with respect to the apo-M11 (2ABO crystal structure) and one can observe the separation between low (blue) and high (green/red) in this low dimensional representation. (**B**) Projection of the holo-M11 simulations (shown in orange dots) onto the subspace spanned by the top three anharmonic modes by the apo- simulations (from the left hand panel) reveal partial overlaps between the apo- and holo-M11 simulations. Notably, the overlaps span areas that have a lower RMSD (blue conformers in the left hand panel) and higher RMSD (green conformers in the left hand panel). (**C**–**E**) Analysis of the motions encoded in the top three anharmonic modes from the apo-M11 simulations show distinct coupling between the motions of α2 to that of the opening/closing of the BH3 domain interaction site. The motions are shown in a movie like representation with darker tones showing the initial state and subsequent frames being shown in lighter tones.

**Figure 4 biomolecules-10-01308-f004:**
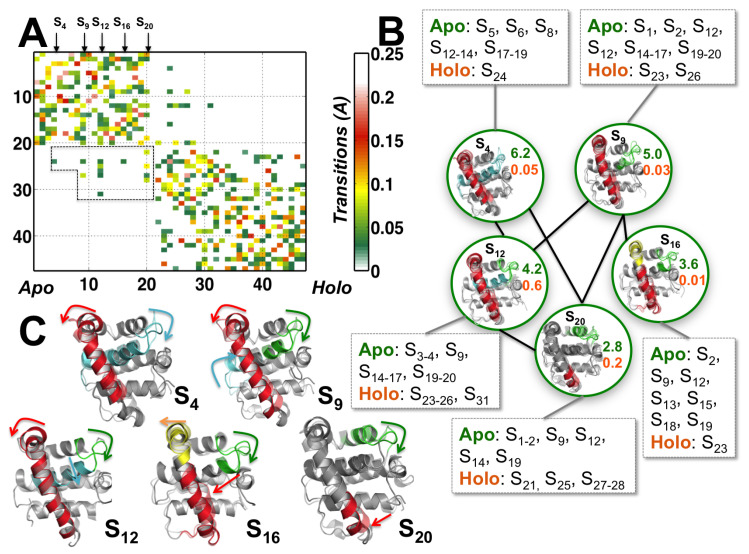
**Conformational substates that act as intermediates between apo- and holo-M11 undergo transient unfolding of α2.** (**A**) Depiction of the conformational substate hierarchy (level 3) summarizing substates from apo- to holo- states from Figure 3, along with the transitions quantified by the affinity, i.e., structural and dynamical similarity between apo- and holo-M11 conformers in each substate. For clarity, we have suppressed the holo- to holo- and apo- to apo-M11 transitions (self-transitions). Arrows along the substates indicate the apo-M11 conformational substates that transit into the holo-M11 simulations representing intermediate states. Transitions with an affinity value of less than 0.01 and greater than 0.2 are shown in white. Transition between apo- and holo M11 states are shown in a dotted box. (**B**) Structural representation of the transitions between apo- and holo-M11 simulations. Five sub-states are represented as nodes of a network (shown in green circles), and, for clarity, the network is expanded to include both apo- and holo- substates as nodes that are directly connected to these intermediate states. For each node, a random sampling of conformers that belong to each sub-state is shown. S${}_{4,9,12,16,20}$ represent apo-states that eventually transit into holo states; the cartoon representation shows the structural details of the apo- and holo-M11 conformers. The numbers accompanying each node in green and orange indicate the relative population of apo- and holo-M11 conformers respectively in that substate. (**C**) Structural changes mediating the conformational changes between apo- (solid cartoons) and holo- (transparent) states summarized by arrows.

**Figure 5 biomolecules-10-01308-f005:**
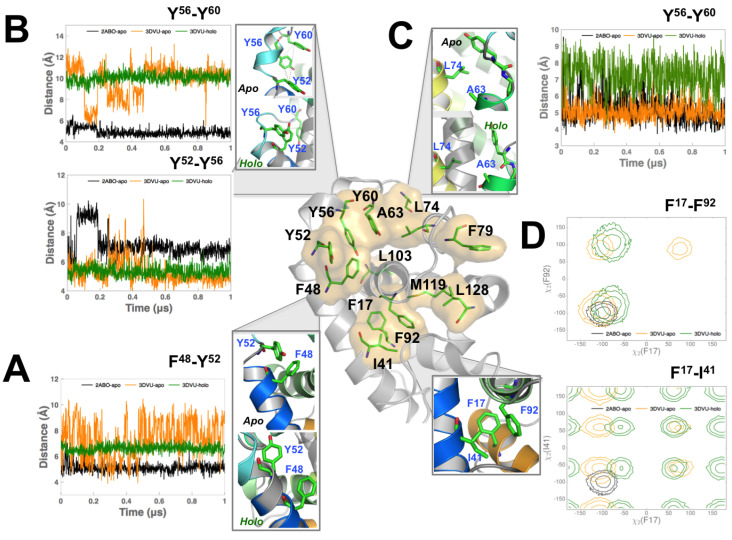
**Mutations to Viral BCL2 protein (vBCL2) residues far away from binding groove significantly impact binding of BECN1 BH3 domain.** (**A**) Distance variation of Phe48 with Tyr52 in the 2ABO-apo (black), 3DVU-apo (orange), and 3DVU-holo (green) simulations. Note that the increase in the flexibility for Phe48 when BECN1 BH3D is removed from the binding site. (**B**) Distance fluctuations of the C${}^{\beta }$ atoms of Tyr52, Tyr56 and Tyr60 for the three simulations. While Tyr52-Tyr56 interaction in the 3DVU-apo simulation remains stable, a small segment of the simulation (from 0.2–0.4 μs) indicates a partial unfolding in this region, sampling distances observed in the 2ABO-apo simulations. Tyr56-Tyr60 interaction on the other hand is stable in both the apo- and holo-M11 simulations, but undergoes dynamic fluctuations in the 3DVU-apo simulations. (**C**) Distance variation of Ala63 and its hydrophobic interactions with Leu74. Observe that upon binding BECN1 BH3D, the fluctuations in the distance between the C${}^{\beta }$ atoms of Ala63 and Leu74 increase when compared to the 2ABO-apo and 3DVU-apo simulations. (**D**) Hydrophobic interactions between Phe17, Phe92, and Ile41 quantified by the $\chi_{2}$ angle distributions of their respective side-chains. Up to three standard deviation intervals from the mean value of the $\chi_{2}$ values are depicted using respective color schemes that are described for the three simulations. For the Phe17-Ile41 side chain interaction, note that the 2ABO-apo simulations only sample a narrow range of $\chi_{2}$ angles; the 3DVU-holo and 3DVU-apo simulations sample additional states not seen in the 2ABO-apo simulations. For the Phe17-Phe92 interaction, there is overlap between the apo- and holo-M11 simulations, but the 3DVU-apo and 3DVU-holo simulations sample additional states not seen by the 2ABO-apo simulations.

**Table 1 biomolecules-10-01308-t001:** Mutations to residues far away from BH3 domain binding site impact vBcl2 function. ${}^{†}$ refers to the mutation that destabilizes the protein (see text below) and, hence, no Isothermal Titration Calorimetry (ITC) measurements are available.

	$K_{d}$ (μM)	$K_{a}$ (1/M)	ΔH (kJ/mol)	ΔG (kJ/mol)	ΔS (J/K·mol)	Distance (Å)
WT	0.84±0.0	$120.0\pm 24\times 10^{4}$	−53.6±3.9	−33.4±0.04	−70.5±13.6	-
F48A	63.4±2.05	$15.8\pm 0.5\times 10^{4}$	−36.1±3.5	−23.1±0.07	−45.4±12.7	9.4
A63K	6.5±0.04	$16.0\pm 0.9\times 10^{4}$	−39.7±0.45	−28.5±0.02	−38.9±1.60	11.9
Y56A	3.38±0.4	$30.0\pm 3\times 10^{4}$	−37.8±7.9	−30.1±0.2	−27.0±28.5	9.7
F17A ${}^{†}$	N/A	N/A	N/A	N/A	N/A	19.7

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
