# Peer review of "Transient Unfolding and Long-Range Interactions in Viral BCL2 M11 Enable Binding to the BECN1 BH3 Domain"

_biomolecules, 2020, doi:10.3390/biom10091308_

Round 1

Reviewer 1 Report

The manuscript explores the conformational changes in protein (vBLC2s) when binds to BH3Ds.

This study is the continuation of other articles by a part of the authors on the same system. It has been carefully planned, the theoretical methodology used is perfectly valid and the conclusions obtained are subsequently validated with some experimental results.

Some minor concerns:

  • In the Methods section, authors use the AMBER version 14.0, however in the Results section is AMBER 12.
  • Page 3. Line 48. Fig. s 1C and 1D should be Figs. 1C and 1D.
  • Table 1. A legend is missing for F17A†

Author Response

The manuscript explores the conformational changes in protein (vBLC2s) when binds to BH3Ds. This study is the continuation of other articles by a part of the authors on the same system. It has been carefully planned, the theoretical methodology used is perfectly valid and the conclusions obtained are subsequently validated with some experimental results.

We thank the reviewers for their comments to help improve our manuscript.

Some minor concerns:

  • In the Methods section, authors use the AMBER version 14.0, however in the Results section is AMBER 12.

We have corrected the same.

  • Page 3. Line 48. Fig. s 1C and 1D should be Figs. 1C and 1D.

Corrected. 

  • Table 1. A legend is missing for F17A†

Done. We have added the legend in table 2 (we inserted the additional table based on reviewer 2's comments). 

Reviewer 2 Report

This manuscript from Ramanathan and colleagues reports molecular dynamics simulations of the murine γ-herpesvirus 68 vBCL2 (M11) in the free form as well as when bound to the BH3 domain (BH3D) to reveal the associated conformational changes. Furthermore, the authors also identify, and later verify by mutagenesis, a network of long-range hydrophobic interactions that can affect BH3D binding. I have a few general and specific comments for consideration by authors:

  • Is it not feasible to carry out an MD simulation of 2ABO-apo as the starting structure in the presence of BH3D (just placed within the simulation box) to study the mechanism and conformational changes associated with the complex formation? Such a starting input will be closer to the physiological conditions and provide observations directly relevant to drug targeting.
  • In the PDB deposited structure, 2ABO has two additional amino acid residues (Glu135-Asp136) at the C-terminal end compared to the 3DVU structure. Similarly, there is an additional residue (Lys5) at the 3DVU N-terminal end. How are these differences normalized in the MD simulation reported here?
  • The interactions between the vBCL2 and BH3D should be included in more details (at residue level), specifically when presenting the simulations results of 3DVU (apo or holo).

Specific comments:

  • Lines 191-192: Table 1 shows the ITC quantification and not the three systems used for MD simulations.
  • Lines 219-221: It appears that the larger fluctuations are in or near loop regions that likely represent destabilized conformations once the bound BH3D is taken out from the complex.
  • Line 224: As illustrated in SI Fig. “S3”
  • Lines 226-227: What is the plausible mechanism for the helix-to-coil transition of residues 33-37 in the 2ABO-apo simulations?
  • Lines 352-354: Similarly, it is noted that Phe48 does not show stable interactions with the Tyr52 in the 3DVU-apo simulations but no associated reasoning is provided. Furthermore, it seems intuitive that 3DVU-apo behavior should be similar to that of 2DVU-apo unless binding of BH3D led to “irreversible” conformational changes that could not be overcome (i.e. to transition from 3DVU-apo to 2ABO-apo) within the MD simulation runs.
  • Line 415: There is no Supplementary Table S1.
  • Lines 474-476: Authors should include a preliminary sequence/structural analysis to indicate the level of conservation for the vBCL2 residues that form the proposed network of hydrophobic interactions.
  • Fig 1C: label the three Tyrosine residues.
  • Figure 1D: Improve the figure for visual clarity for showing the conformational changes in tyrosine residues.
  • Figure 2A legend: 2ABO-apo is shown in the “green” line.
  • Figure 2A: Label Panels E and F.

Author Response

This manuscript from Ramanathan and colleagues reports molecular dynamics simulations of the murine γ-herpesvirus 68 vBCL2 (M11) in the free form as well as when bound to the BH3 domain (BH3D) to reveal the associated conformational changes. Furthermore, the authors also identify, and later verify by mutagenesis, a network of long-range hydrophobic interactions that can affect BH3D binding. I have a few general and specific comments for consideration by authors:

We thank the reviewer for their insightful comments.

Is it not feasible to carry out an MD simulation of 2ABO-apo as the starting structure in the presence of BH3D (just placed within the simulation box) to study the mechanism and conformational changes associated with the complex formation? Such a starting input will be closer to the physiological conditions and provide observations directly relevant to drug targeting.

This is a great suggestion, one that we had initially considered as part of our simulation set-up. Although we were aware of similar studies for studying the mechanisms of other protein-protein interactions (Jost Lopez, A. et al 2020), we noted that the size of the simulation box and solvation conditions could have a pronounced effect in terms of drawing scientific conclusions. More recent papers (Gapsys and de Groot, eLife 2020) also highlight the importance of considering box width and volume for accurate sampling and estimating kinetics and other relevant parameters. Given that the conformational changes for the BH3D in the presence of the viral BCL2 takes place in milliseconds timescales, we plan to study these with some of our adaptive sampling techniques as part of a future study.

References:

  1. Jost Lopez, A., et al. (2020). "Quantifying Protein–Protein Interactions in Molecular Simulations." The Journal of Physical Chemistry B 124(23): 4673-4685.
  2. V. Gapsys, B. L. de Groot, On the importance of statistics in molecular simulations for thermodynamics, kinetics and simulation box size. eLife: https://elifesciences.org/articles/57589.

In the PDB deposited structure, 2ABO has two additional amino acid residues (Glu135-Asp136) at the C-terminal end compared to the 3DVU structure. Similarly, there is an additional residue (Lys5) at the 3DVU N-terminal end. How are these differences normalized in the MD simulation reported here?

We thank the reviewer for picking up on this. While the structures for MD simulations were essentially modeled with the residues that were in the original crystal structure, for analyses purposes, we did not include them. This decision to not include the residues in the analyses is primarily driven by the fact that the N- and C-terminal ends are capped, and the terminal ends of the protein are not known to influence the BECN1 BH3D binding to the binding site. We have made this explicit in our manuscript (page 3: lines 75-80).

The interactions between the vBCL2 and BH3D should be included in more details (at residue level), specifically when presenting the simulations results of 3DVU (apo or holo).

We thank the reviewer for this suggestion and agree with the reviewer that the analysis of interactions between vBCL2 and BH3D is important. Indeed, we have previously described the direct interactions between M11 and the BECN1 BH3D in substantial detail, including the impact of mutating these residues on binding of the two proteins (Refs. 13 and 19 in the main text of the paper). Therefore we have not detailed these interactions in this manuscript. Our previously published studies that detail this interaction are now mentioned and appropriately referenced in the introduction (Pages 1 and 2: lines 42-44). Further, we note here that the current study tests our hypothesis that there are long-range interactions in vBCL2, which has not been previously investigated. Therefore, we have primarily chosen to focus on the conformational changes induced within vBCL2 (rather than focus on the BH3 domains) to demonstrate that such long-range interactions do exist and mutations farther away from the binding site can induce significant changes in binding. In future studies, we intend to further explore how BH3Ds bind to vBCL2 -- since the binding mechanism may be altered when diverse BH3 domains undergo folding upon binding to the vBCL2, with different affinities dictated by the interaction of interface residues.

Specific comments:

  • Lines 191-192: Table 1 shows the ITC quantification and not the three systems used for MD simulations.

This was an unintended error. Table references are corrected. 

  • Lines 219-221: It appears that the larger fluctuations are in or near loop regions that likely represent destabilized conformations once the bound BH3D is taken out from the complex.

We have clarified the language in the paper to reflect the same (lines page 8: 230-231).

  • Line 224: As illustrated in SI Fig. “S3”

We have corrected the information in the supporting information.

  • Lines 226-227: What is the plausible mechanism for the helix-to-coil transition of residues 33-37 in the 2ABO-apo simulations?

We have added a couple of sentences to clarify the possible mechanism of this helix-to-coil transition (Page 8: lines 237-242). 

  • Lines 352-354: Similarly, it is noted that Phe48 does not show stable interactions with the Tyr52 in the 3DVU-apo simulations but no associated reasoning is provided. Furthermore, it seems intuitive that 3DVU-apo behavior should be similar to that of 2DVU-apo unless binding of BH3D led to “irreversible” conformational changes that could not be overcome (i.e. to transition from 3DVU-apo to 2ABO-apo) within the MD simulation runs.

We have added a sentence about the plausible mechanism by which Tyr52 and its interaction with specific residues across BCL2 may induce such conformational changes upon BH3D removal/ binding (page 12-367-373).

  • Line 415: There is no Supplementary Table S1.

We have included supplementary table S1 in the manuscript.

  • Lines 474-476: Authors should include a preliminary sequence/structural analysis to indicate the level of conservation for the vBCL2 residues that form the proposed network of hydrophobic interactions.

We thank the reviewer for pointing this out. We have added supporting information text S4, Figure S6 (depicting the sequence alignment to show low sequence similarity) and Figure S7 (depicting the structure based representation of the hydrophobic interaction network) to clarify this.

  • Fig 1C: label the three Tyrosine residues.

Done.

  • Figure 1D: Improve the figure for visual clarity for showing the conformational changes in tyrosine residues.

Done. We have also modified the figure to improve its clarity and distinguish between the apo- and holo-state conformations with explicit labels.

  • Figure 2A legend: 2ABO-apo is shown in the “green” line.

Done.

  • Figure 2A: Label Panels E and F.

Done. We have corrected it to clarify the “zoomed in” version from panel C.

Round 2

Reviewer 2 Report

The overall responses from the authors are satisfactory. I recommend the revised manuscript for publication.

As a minor suggestion: A brief note along the lines of the authors response regarding the "MD simulation of 2ABO-apo as the starting structure in the presence of BH3D" could be incorporated in the Methods and/or the Discussion section.

Author Response

The overall responses from the authors are satisfactory. I recommend the revised manuscript for publication.

As a minor suggestion: A brief note along the lines of the authors response regarding the "MD simulation of 2ABO-apo as the starting structure in the presence of BH3D" could be incorporated in the Methods and/or the Discussion section.

We thank the reviewer for their comments. We have added a description of the simulating the 2ABO-apo starting structure in the presence of the BH3D in the discussions section, along with additional references (Page 16: lines 494-504).